

# Toward coherent space-time mapping of seagrass cover from satellite data: example of a Mediterranean lagoon

Guillaume Goodwin[1], Marco Marani[2], Sonia Silvestri[3], Luca Carniello[2], and Andrea D'Alpaos[4]

[1]Fish-Pass Environnement, 18, rue de la Plaine, 35830 Laillé, France
[2]DICEA, Universita di Padova, via Marzolo, 9, Padova, Italy
[3]Dipartimento di Scienze Biologiche, Geologiche e Ambientali, Alma Mater Studiorum Università di Bologna, Via S. Alberto 163, Ravenna, Italy
[4]Dipartimento di Geoscienze, Universita di Padova, via Gradenigo, 6, Padova, Italy

**Correspondence:** Guillaume Goodwin (willgoodwin1201@gmail.com)

**Abstract.** Seagrass meadows are a highly productive and economically important shallow coastal habitat. Their sensitivity to natural and anthropogenic disturbances, combined with their importance for local biodiversity, carbon stocks and sediment dynamics, motivate a frequent monitoring of their distribution. However, generating time-series of seagrass cover from field observations is costly, and mapping methods based on remote sensing require restrictive conditions on seabed visibility, lim-

iting the frequency of observations. In this contribution, we examine the effect of accounting for environmental factors such as the bathymetry and median grain size ($D_{50}$) of the substrate, as well as the coordinates of known seagrass patches, on the performance of a Random Forest (RF) classifier used to determine seagrass cover. Using 148 Landsat images of the Venice Lagoon (Italy) between 1999 and 2020, we trained a RF classifier with only spectral features from Landsat images and seagrass surveys, respectively from 2002 and 2017. Then, by adding the features above and applying a time-based correction on

predictions, we created multiple RF models with different feature combinations. We tested the quality of the resulting seagrass cover predictions from each model against field surveys, showing that bathymetry, $D_{50}$ and coordinates of known patches exert an influence that is dependant on the training Landsat image and seagrass survey chosen. In models trained on a survey from 2017, where using only spectral features causes predictions to overestimate seagrass surface area, no significant change in model performance was observed. Conversely, in models trained on a survey from 2002, the addition of the out-of-image

features and particularly coordinates of known vegetated patches greatly improves the predictive capacity of the model, while still allowing the detection of seagrass beds absent in the reference field survey. Applying a time-based correction eliminates small temporal variations in predictions, improving predictions that performed well before correction. We conclude that accounting for the coordinates of known seagrass patches, together with applying a time-based correction, has the most potential to produce reliable frequent predictions of seagrass cover. While this case study alone is insufficient to explain how geographic

location information influences the classification process, we suggest that it is linked to the inherent spatial auto-correlation of seagrass meadow distribution. In the interest of improving remote sensing classification and particularly to develop our capacity to map vegetation across time, we identify this phenomenon as warranting further research.



# 1 Introduction

Seagrass meadows are emblematic shallow water ecosystems, well-known for their diverse wildlife Sfriso et al. (2001) and capacity to sequester carbon and nutrients Greiner et al. (2013); Russell et al. (2013); Johnson et al. (2017): early landmark valuations of ecosystem services estimated the benefits generated by seagrass and algae beds at over $19,000\,USD(1997){\cdot}ha^{-1}{\cdot}$ $yr^{-1}$, second only to forested wetlands such as swamps and floodplains Costanza et al. (1997). Furthermore, seagrass meadows modify velocity and turbulence regimes Hendriks et al. (2008); Ganthy et al. (2015); Carniello et al. (2016), resuspension

Widdows et al. (2008); Volpe et al. (2011); Hansen and Reidenbach (2013); Carniello et al. (2014); Venier et al. (2011), and sediment trapping Hendriks et al. (2010), influencing sediment dynamics in estuaries and lagoons. Hence, in the current context of rising sea levels Nicholls et al. (2021) and sediment deprivation (Syvitski and Kettner, 2011), understanding their role in shaping coastal landforms is crucial: reliable and reproducible observations of space-time seagrass presence and change are a key missing element towards a complete understanding of tidal environment dynamics, now largely focusing on salt marsh

eco-geomorphodynamics (D'Alpaos et al., 2007; Marani et al., 2007, 2011, 2010; Yousefi Lalimi et al., 2020).

-0cm



**Table 1.** Compilation of published works on seagrass detection from satellite data. Numbered references are referenced in Table A1.

| Sensor | Pixel size | Scenes | Area [km$^2$] | Method | Performance | Ref |
|---|---|---|---|---|---|---|
| Landsat 5 | 30 m | 1 | 200 | Minimum-distance-to-means | UA:11-55%, OA:35% | 1 |
| | 30 m | 1 | 105 | Maximum Likelihood | UA:83-99%, OA:92% | 2 |
| Landsat 4,5,7 | 30 m | 60 | 200 | Multi resolution Segmentation | OA:52-80% | 3 |
| Landsat 5,7 | 30 m | 4 | 94 | Maximum Likelihood | OA:54-100% | 4 |
| | 30 m | 40 | $> 10^6$ | Maximum Likelihood | UA:0-88%, OA:45-85% | 5 |
| Landsat 5,7,8 | 30 m | 49 | 6 | ROI growth | UA:84-92% OA:91-96% | 6 |
| Landsat 5, 8 | 30 m | 2 | 42.6 | Maximum Likelihood | A:89-93%, OA:85% | 7 |
| Landsat 8 OLI | 30 m | 2 | 33,000 | Lyons et al. (2012) | UA:13-95%, OA:29-99% | 8 |
| | 30 m | 1 | 100 | Roelfsema et al. (2014) | UA:22-73%, OA:59% | 9 |
| Landsat 8, Sentinel-2 | 10-30 m | 6 | 140 | Maximum Likelihood | UA:13-95%, OA:65-75% | 10 |
| ALI | 30 m | 1 | 105 | Maximum Likelihood | UA:86-99%, OA:95% | 11 |
| Hyperion | 30 m | 1 | 105 | Maximum Likelihood | UA:89-99%, OA:96% | 12 |
| Sentinel-2 | 10 m | 1 | 100 | Roelfsema et al. (2014) | UA:21-81%, OA:57% | 13 |
| | 10 m | 1-2 | 340 | Empirical | RMSE:14% of area | 14 |
| | 10 m | 1 | 41,000 | Support Vector Machines | UA:up to 99%, OA:72% | 15 |
| Ziyuan-3A | 5 m | 1 | 100 | Roelfsema et al. (2014) | UA:28-67%, OA:54% | 16 |
| CASI-2 | 4 m | 1 | 200 | Minimum-distance-to-means | UA:16-68%, OA:46% | 17 |
| IKONOS, WV2, QB2 | 2.4-4 m | 9 | 142 | hierarchical OBIA | UA:10-57%, OA:52+-4% | 18 |
| QuickBird | 2.4 m | 1 | 200 | Minimum-distance-to-means | UA:9-52%, OA:31% | 19 |
| WorldView-2 | 2.4 m | 1 | 8.6 | SVM /Random Forest | OA:72-94% | 20 |
| WorldView-3 | 2 m | 1 | 100 | Roelfsema et al. (2014) | UA:29-76 %, OA:59% | 21 |
| IKONOS | 2 m | 1 | 1.78 | KM/MD/ML/LSU | UA:65-85%, OA:40-67% | 22 |
| GeoEye1 | 1.65 m | 1 | 20 | ML | UA:78-90%, OA:85.7% | 23 |

Multiple monitoring campaigns, at several different sites and using diverse methods, have been conducted over the years to map seagrass cover, leading to the recent compilation of a global seagrass distribution assessment McKenzie et al. (2020). Field mapping is widely employed to determine vegetation characteristics such as stem density, biomass, metabolism, etc. (e.g. Smith et al. (1988); Caffrey and Kemp (1991); Kutser et al. (2007)), but high costs and long completion times prevent frequent surveys of the state and extent of submerged vegetation. And yet timeliness is particularly important in understanding the response of seagrass meadows to environmental stressors. In favourable conditions and in the growing season, seagrass can recover from heat waves Pedersen et al. (2016); Gamain et al. (2018) or shallow scouring within just a few months Collier and Waycott (2014), making the effect of such disturbances invisible to infrequent observations. Remote sensing using multi- and






hyper-spectral sensors constitutes an attractive alternative and complements field mapping, provided that detection methods can reliably classify the seabed. Such methods have been applied successfully to satellite data for salt marsh vegetation in the Venice Lagoon (Wang et al., 2007; Yang et al., 2020) as well as for the quantification of suspended sediment concentration (Volpe et al., 2011; Zhou et al., 2017); detecting seagrass using the same data sources, remains challenging, due to to highly variable water depth and constituents, but would allow for consistent environmental monitoring. Satellite-born sensors also

provide up to daily observations, and are cost-efficient, making them an ideal support for high-frequency and spatially-extended monitoring.

Table 1 reviews 16 publications concerning seagrass mapping from satellite imagery, ordered by decreasing pixel footprint of the image product. These studies classify seagrass density in steps of 25% pixel surface cover, using various methods such as the Maximum Likelihood method (Cam, 1990), trained Support Vector Machines (Noble, 2006), and Random Forest (Biau and

Scornet, 2016) methods. Landsat data, with 30 m pixels that can be larger than some seagrass patches and few spectral bands, do not perform significantly worse than commercially-available data with higher resolution and a larger number of bands, such as IKONOS. The wider range of performances for Landsat over other products may be a result of the greater number of studies that use this support. Overall, trained machine learning methods perform marginally better than more traditional classifiers on remote sensing data of the same resolution. However, a systematic comparison among different classifiers can

hardly be inferred from the literature, because of the widely different resolutions explored in the existing studies. Indeed, the Maximum Likelihood classifier is used primarily on Landsat products. The wide variety of performances shown in such applications suggests a strong dependence of classification results on the quality of the data and the conditions in which the data were acquired. Indeed, atmospheric conditions, water depth, the presence of waves, as well as chlorophyll and sediment concentrations, all affect reflectance at the water surface, and thus the visibility of the seabed. As a result, acquisitions to be

classified must be subject to a strict selection process, making frequent and regular monitoring difficult.

An inconsistent classification performance, and the consequent irregular monitoring frequency (e.g. see Table 1), poses significant limitations to seagrass cover monitoring: the high primary productivity of seagrass implies that meadow density and canopy characteristics, and therefore spectral reflectance, greatly vary seasonally, with growth stages, and stochastically, with storm-induced thinning or scouring Sfriso and Francesco Ghetti (1998). This high rate of variation in density and extent also

implies that seagrass maps separated by more than a few months cannot capture seasonal, and much less stochastic, variations in seagrass cover. Yet few of the studies listed in Table 1 classify more than one image (Lyons et al., 2012; Dekker et al., 2005; Wabnitz et al., 2008; Hossain et al., 2015; Kohlus et al., 2020; Roelfsema et al., 2014), while most limit the analyses to single illustrative data acquisitions.

Here we contribute to the development of a reliable method for seagrass space-time mapping by developing a modified

Random Forest (RF) classifier Bakirman and Gumusay (2020), and by applying it to determine seagrass cover in approximately 150 Landsat scenes between 1999 and 2020 from the Venice Lagoon, Italy. Based on 9 field surveys performed between 2002 and 2017, we investigate the influence of environmental conditions, known seagrass coordinates, and temporal persistence of detected features on the performance of the classifier.



## 2 Materials and method

With a surface area of $550\ km^2$, the Venice Lagoon is the largest lagoon in the Mediterranean Sea. Due to its socio-economic and environmental importance, and to a significant erosional trend (Carniello et al., 2009), the ecological and morphological state of the lagoon has been systematically monitored for decades. Historic and modern bathymetric information complemented by a long record of tidal data provide the opportunity of quantitatively describing, also via numerical modeling, the detailed hydrodynamic circulation throughout the lagoon Tommasini et al. (2019). Numerous sampling campaigns of sediment

composition on the lagoon bed Amos et al. (2004); Guerzoni and Tagliapietra (2006); Carniello et al. (2012) are used to infer the preferred habitat of seagrass meadows, and multiple submerged vegetation surveys conducted throughout the 21$^{\text{st}}$ century provide a reference for classification testing.





**Figure 1.** Out-of-image data used as features for seagrass detection. Background image: gridded bathymetry of the Venice Lagoon, collected in 2003 and updated only for the inlets in 2012; data points: gridded median grain size. Triangles indicate the locations of tide and wind gauges. The greyed area represents a zone in the lagoon where no seagrass was found during field surveys. The coordinate system used is in the local Monte Mario projection (EPSG: 3003). Inset image: ©Google Earth





With this wealth of data, the Venice Lagoon is a prime candidate to test new approaches of submerged vegetation detection. In this section, we first describe the collection and processing of local data as well as satellite images from Landsat 7 ETM and Landsat 8 OLI repositories, used as input and for validation. Then, we describe the structure of the Random Forest model developed to classify the images, and explain how local environmental data, as well as data pertaining to the coordinates of known seagrass patches, are added to improve the skill of the classifier. We also describe a simple process used to correct initial predictions, based on the analysis of the predicted seagrass cover time-series. Finally, we define the metrics used to assess the classifier's performance.

## 2.1 Local data collection and processing

A full lagoon bathymetric survey was performed in 2003 by the Venice Water Authority (Nuova-Technital, 2007) and subsequently updated to account for engineering works that modified the bathymetry within the inlets of Chioggia, Malamocco, and Lido (Carniello et al., 2009) (Figure 1). Furthermore, samples of bed sediment composition have been collected over the years, resulting in a dataset of more than 900 estimates of median grain size ($D_{50}$) (Carniello et al., 2012) (Figure 1). A network of tidal and wind speed gauges records hourly water surface elevation, wind speed, and direction in key locations of the lagoon since before 2000: among these, we chose stations close to Chioggia in the South and the Saline station in the North (red triangles in Figure 1, source data: (https://www.comune.venezia.it/content/dati-dalle-stazioni-rilevamento)). Those stations were chosen as they have the most consistent record over the 1999-2020 period. The station in Saline records both wind velocities and tidal elevation, whereas these data are recorded by two separate stations, located $500m$ apart, in Chioggia.

In addition to physical data, we used field surveys (performed by SELC, https://www.selc.it/) detailing the extent of seagrass cover, species composition, and stem density, hosted on the Atlante della Laguna website (cigno.atlantedellalaguna.it/maps/6/view). These surveys were conducted across the entire lagoon in the late summer of 2002, 2004, 2009, 2010 and 2017, by observation from a boat in the field, with support from aerial images, which allows a fine delineation of seagrass patches. However, given the extent of the lagoon, field surveys have taken place over periods of several months, such that they do not represent the state of the lagoon at a single moment in time. Furthermore, the criteria used to estimate seagrass cover may have varied over time or as a result of different observers. Additional surveys were conducted in the late summer of every year between 2006 and 2015, with density classes derived from field, satellite and aerial image observations. In this study, we do not use a discretization into density classes and instead consider all density classes that are not bare as vegetated. The footprint of these additional surveys focuses on the inlets of Lido, Malamocco and Chioggia, as seen in Figure A1. Finally, we digitised a specific patch of seagrass near Chioggia in 25 Landsat images between 2000 and 2020, also shown in Figure A1.

## 2.2 Satellite data collection and processing

We downloaded 164 cloud-free Level 2 multispectral acquisitions from the Landsat 7 ETM and Landsat 8 OLI data repository, covering the entire Venice Lagoon between 1999 and 2020. Landsat 5 data were not considered due to inconsistent results in the studies examined in Table 1. Level 2 products are atmospherically corrected using the LEDAPS (Schmidt et al., 2013) and LaSRC algorithms (Ilori et al., 2019), and yield surface reflectance values corrected for the scattering and absorbing effects





of gas, vapour, and aerosols. In May 2003, the Scan Line Correction (SLC) system on the Landsat 7 ETM sensor failed, causing all subsequent ETM scenes to contain strips of empty data. Nevertheless, we did not disregard these acquisitions, and instead have designed our methods to account for missing data. From July 2013 onward, Landsat 8 OLI data were brought online, effectively resolving the issue for the purposes of our study. We selected a number of images among those downloaded

according to two criteria (Figure 2): a) the tidal elevation at the full hour closest to overpass time is less than $0.75\ m$ above the national datum (Rete Altimetrica dello Stato ,1897). This limits the effect of water column absorption on visibility of the lagoon bed; b) the third quartile of the series of wind speeds up to three days prior to overpass time does not exceed $8\ m \cdot s^{-1}$ and the 90-th percentile does not exceed $15\ m \cdot s^{-1}$. This criterion limits the probability of inorganic suspended sediment and wind-waves impacting visibility of the seabed. Applying these criteria leaves 148 scenes that are *a priori* fit for the detection

of seagrass meadows on the lagoon bed. Landsat scenes corresponding to field surveys are listed in Table A2, while those associated with digitised reference information are shown in Table A3.

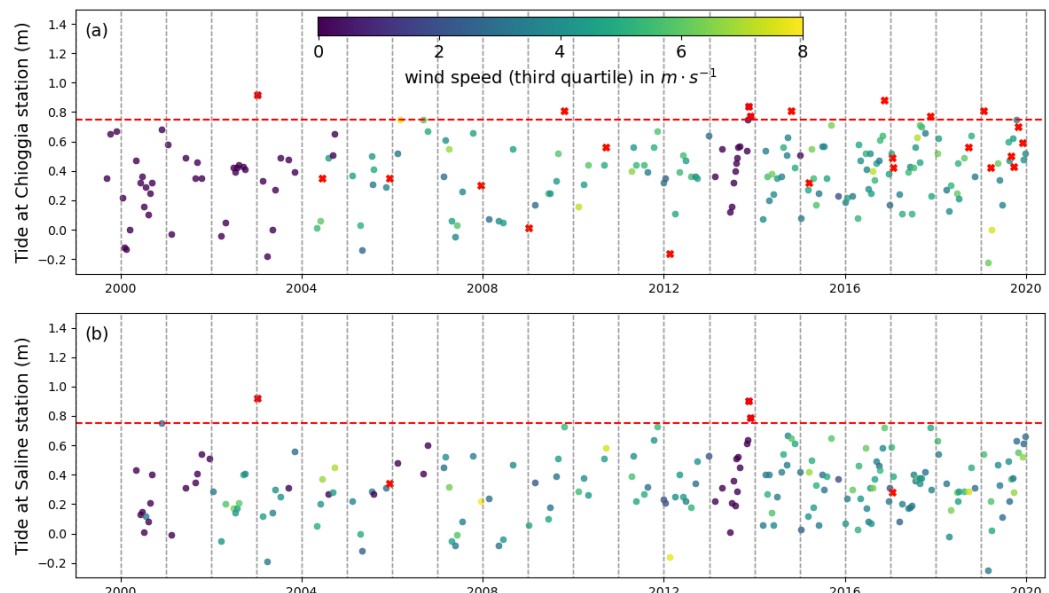

**Figure 2.** Tide and wind conditions used to select Landsat scenes at the stations of (a) Chioggia and (b) Saline. The red dotted line shows the high limit for tidal elevation relative to station datum: scenes where water level exceeds this elevation were not selected and are shown in red. Scenes where wind speed exceeded the limit condition are also shown in red.

Bathymetric (Carniello et al., 2009) and sediment grain size data (Carniello et al., 2012), excluding channels and salt marshes, were gridded with the Geospatial Data Abstraction Library (GDAL) (GDAL/OGR contributors, 2021) at a pixel size of $30\ m$ according to the nearest neighbour method. All surveyed and digitised seagrass meadows, initially in vector for-

mat, were also rasterised using GDAL at a pixel size of $30\ m$, to match bathymetry and sediment size data. Seagrass cover





vector data were gridded by considering pixels covered by a seagrass polygon by more than 50% as vegetated (coded as "1"). The remaining pixels were considered bare soil (coded as "0").

## 2.3 Description of the Random Forest model

We developed a Random Forest (RF) classifier designed to identify the presence of seagrass meadows on the lagoon bed

based both on spectral and non-spectral information. Random Forest modelling is an ensemble learning algorithm that uses the results of a large number of decision trees (Ho, 1995). This class of algorithms is being used more frequently in remote sensing classification problems in general (Pal, 2005; Belgiu and Drăgu, 2016) and in seagrass detection in particular (Table 1). RF classifiers are trained to predict a set of target properties based on the values of a set of several features. Here, we train multiple classifiers to predict the absence (0) or presence (1) of seagrass, each with a different set of features, and using different training

target values.

Common features used to determine seagrass cover are the atmospherically corrected spectral reflectance values in the blue, red, green and sometimes near-infrared bands (see references in Table 1). When the seagrass is submerged, these reflectances at the water surface are not linearly connected to the reflectance of the bed. Indeed, water depth, organic and inorganic suspended sediment concentrations, water surface roughness, all contribute to define a complex relation between intrinsic seagrass

reflectance and remote sensing at-water-surface reflectance. These parameters are rarely simultaneously measured or acquired to be used in a radiative transfer model (Lee et al., 1998; Lee and Carder, 2002) to infer the bed reflectance.

Given the uncertainties affecting the spectral reflectance properties that may be retrieved from remote sensing, it is reasonable to leverage all available information with the aim to reliably perform seagrass mapping across multiple acquisitions.





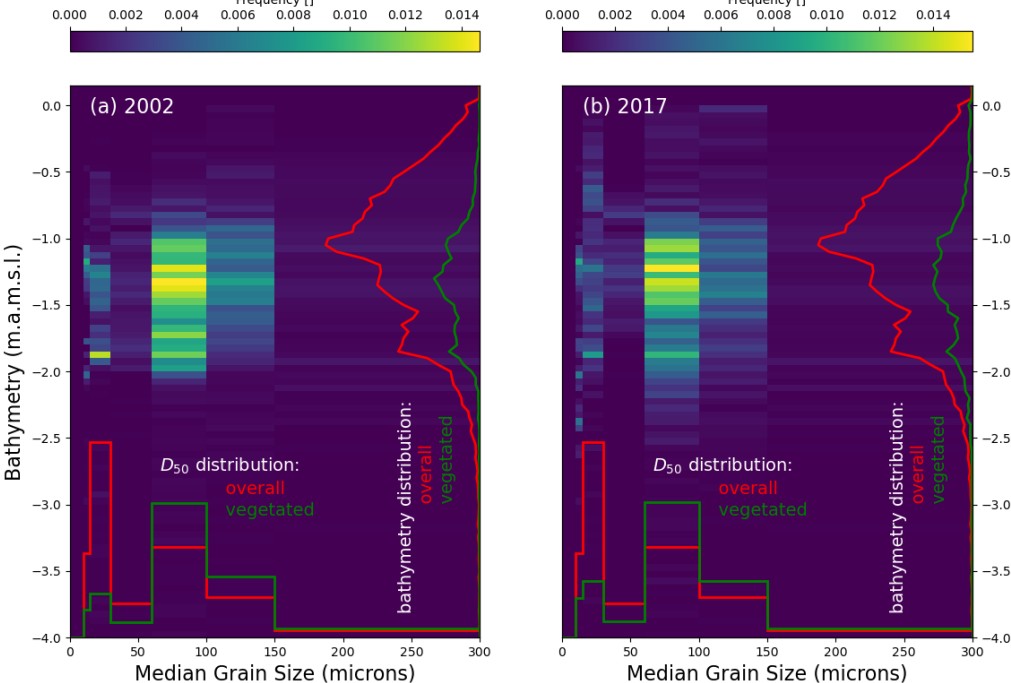

**Figure 3.** 2D frequency distribution of seagrass according to bathymetry relative to IGM reference and median sediment grain size ($D_{50}$). 1D frequency distributions in green are a projection of the 2D distributions on their respective axis. 1D frequency distributions in red are the distributions of bathymetry (y-axis) and $D_{50}$ (x-axis). (a) seagrass distribution for the 2002 survey; (b) seagrass distribution for the 2017 survey.

Figure 3 shows the frequency distribution of seagrass meadows across bathymetry and median sediment grain size ($D_{50}$) for
full-lagoon field surveys performed in 2002 (a) and 2017 (b). We notice that seagrass meadows in the Venice Lagoon occupy a quite characteristic range of bed elevations, mostly between $-2.1m$ and $-0.5m$ above datum, much narrower than the overall bathymetric range. Furthermore, the peak seagrass occurrence frequency does not correspond with the mode of the bathymetry, i.e. it is not located at the most commonly occurring bottom depth. This indicates that seagrass occurs within a preferential range of water depths, dictated by their need of access to light and by preferred flooding frequencies Carruthers et al. (2002).
Even more evidently, seagrass is preferentially found on fine sandy seabeds, where $D_{50}$ ranges between $60\mu m$ and $150\mu m$, even though these are not the most common sediment sizes in the lagoon. Whether such ranges of bathymetry and sediment size are purely the expression of preferred habitat or the product of self-organisation and eco-geomorphic feedbacks is not debated in this contribution. However, the existence of a relationship between environmental parameters, such as bathymetry and $D_{50}$, and seagrass distribution may be of assistance to the developing effective algorithms for the detection of seagrass
meadows.




A habitat constraint invariably translates into a constraint in geographical distribution. Figure 4 shows the geographical distribution of seagrass meadows across available field surveys described above, where each pixel expresses the cross-product $\Pi_N$ in Equation 1:

$$\Pi_N = \frac{\sum_{i=0}^{N} \sum_{j \neq i}^{N} V_i \cdot V_j}{N * (N-1)} \tag{1}$$

where N is the number of surveys considered, and $V_i$ is the array representing seagrass cover, with values comprised between 0 and 1. In both the full-lagoon and inlet surveys, large swathes of seagrass meadows harbour $\Pi = 1$ or $\Pi = 0$ (shown as empty data in Figure 4).

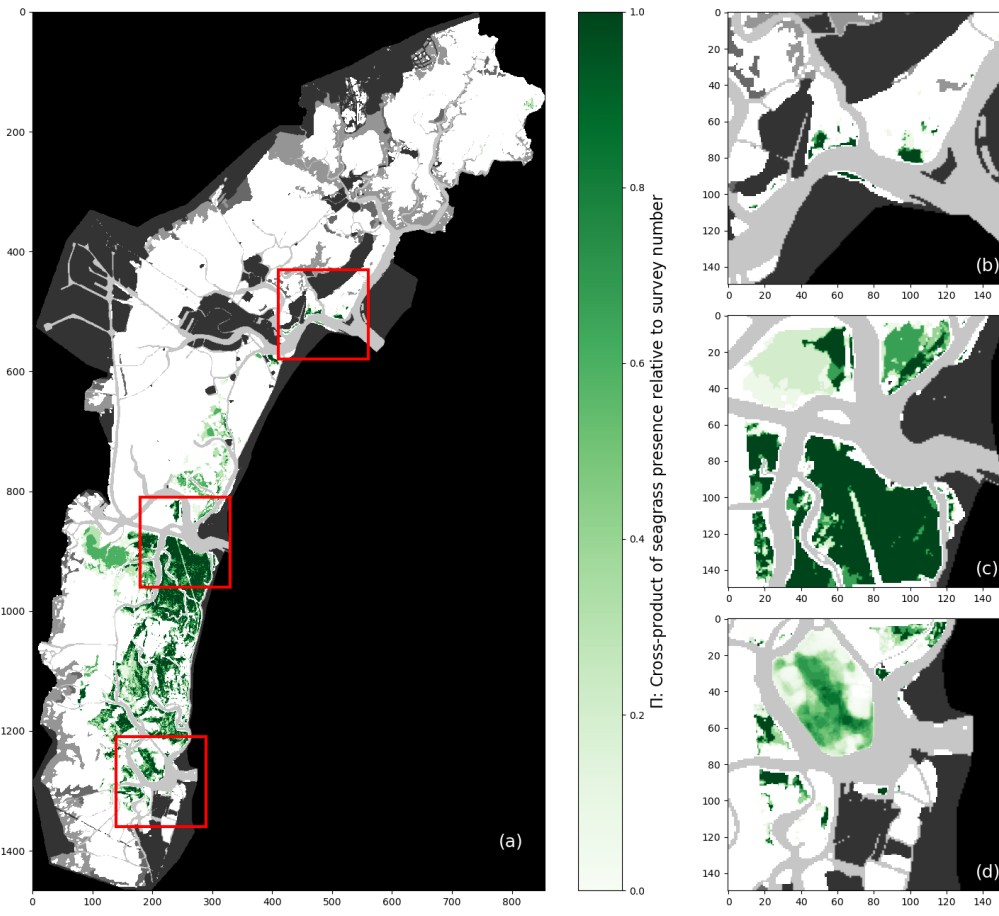

**Figure 4.** Map of $\Pi$ calculated for (a) 5 full-lagoon surveys; (b-d) 6 inlet surveys, respectively at the inlets of Lido (b), Malamocco (c), and Chioggia (d). Additionally, the area appearing blurred in (d) represents the map of $\Pi$ for all surveys plus 25 manually classified images.

4 corroborates the notion that preferred habitats, shown simply in Figure 3, constrain the geographical range of seagrass meadows in the Venice Lagoon. The North and Central lagoon host patchy, disconnected, meadows, while the Southern la-





goon hosts large interconnected meadows separated by navigation canals. This observation implies a degree of spatial auto-correlation in the presence of seagrass meadows, which may be explained by mechanisms of seagrass patch development: these are in general driven by environmental factors, such as differences in depth, nutrient fluxes, and salinity (Ghezzo et al., 2011; Sfriso et al., 2003), while clonal reproduction is the primary mechanism through which seagrass plants become established after an initial colonisation McMahon et al. (2014) and sexual reproduction allows their spread over large distances.

The global Moran Index provides additional evidence of spatial autocorrelation in seagrass distributions. This index represents the degree of spatial auto-correlation (SAC) for a dataset, varying between $-1$ and $1$, with $-1$ corresponding to evenly spaced patches, $0$ being the value approached by a random distribution of elements, and $1$ being attained if a space is divided in two halves of contiguous elements of the same value (Fan and Myint, 2014). This further indicates a strong positive auto-correlation that suggests a zonation of the lagoon in areas, with some being favourable to seagrass development and others

not. Here, all full-lagoon surveys present a global Moran's Index greater than 0.8. As a result, the value of pixel coordinates, expressed as row and columns in the array, may represent a useful feature in seagrass detection. Because such a feature carries the risk of introducing confirmation bias in the results, its influence on prediction variability will be examined closely.

Figures 3 and 4 show that features other than spectral reflectance have potential to improve the performance of a classifier seeking to determine seagrass presence. To assess the impact of their inclusion among predictors on classification uncertainty,

we train RF classifiers with a combination of features comprising of spectral, environmental, and location-based features, as shown in Figure 5. In order to be used simultaneously in the RF classifier, all features used were first normalised relative to the $5th$ and $95th$ percentile of their value in their respective zone (instead of the minimum and maximum). Because seagrass patches in the the North and Central lagoon show different sizes and density than in the Southern lagoon, we divided the lagoon in two geographic zones at the Malamocco Inlet channel (see Figure 4) and adopted a different RF classifier in each

zone. The predictors (features) used are spectral reflectances, bed elevation, and $D_{50}$. The RF classifiers, implemented in the Scikit-Learn package (Pedregosa et al., 2011), include 100 trees, each considering 3 input features with a maximum depth of 30 nodes. This RF classifier structure was chosen to avoid overfitting, which would limit the model's capacity to classify seagrass in the presence of a variable spectral response (caused by the presence of algae, local chlorophyll hotspots, etc.). The RF classifiers are trained using Landsat scenes taken on 14/09/2002 and 30/08/2017, corresponding to field surveys conducted

in the summers of 2002 and 2017. These survey dates were chosen over the remaining ones because the corresponding remote sensing acquisitions are not affected by the Landsat 7 Scan Line Correction (SLC) failure. For each survey, we train one model per each northern/southern geographical zone, and per each combination of features, resulting in 8 RF models being trained in total. We then use these models to predict seagrass cover in 148 selected Landsat images (see Figure 2).

## 2.4 time-based correction

Seagrass cover predictions are liable to instability due to variations in data quality, water absorption and scattering properties. The failure of the Landsat7 ETM scan line corrector on 31/05/2003 caused all subsequent Landsat7 ETM data to contain no-data strips, the position of which varies between images. To account for this discrepancy in data, we gave any no-data pixel at a given classification date $t$ the classification it had in its previous classification $t-1$ and the following classification $t+1$,





provided these were identical. If not, the pixel remains as no-data. Further instability in the predictions may be caused by
classifications switching from bare to vegetated (or inversely) for a single image, for instance because of changes in seagrass or
seabed reflectance caused by the intermittent presence of macroalgae or fishing activities. For instance, a pixel may appear bare
for a set of contiguous scenes, then be classified as vegetated for one scene only, to return to a bare classification thereafter.
Given the frequency of the scenes acquired, we consider it unreasonable to assume for seagrass patches to appear or disappear
in isolated scenes. Indeed, we considered it unlikely for both a scouring event and subsequent full recovery to occur within less
than 5 months, which corresponds to the largest gap between images. Instead, these abrupt changes in seagrass classifications
may be caused by the apparition or disappearance of algal patches, which appear similar to seagrass in the visible and infrared
bands but, lacking a rooted anchor to the seabed, are more mobile than seagrass. Consequently, we applied the following
post-processing correction rule: any pixel being classified in a scene at time $t$ differently from classifications in scenes $t-2$,
$t-1$, $t+1$, and $t+2$ (provided these are identical), is given the opposite classification. Finally, given the resolution of Landsat
images, isolated patches of less than 4 pixels in surface area, whether they be bare or vegetated, are switched to the dominant
classification around them. These corrected sets of predictions represent another set of 8 predictions to validate (see Figure 5).

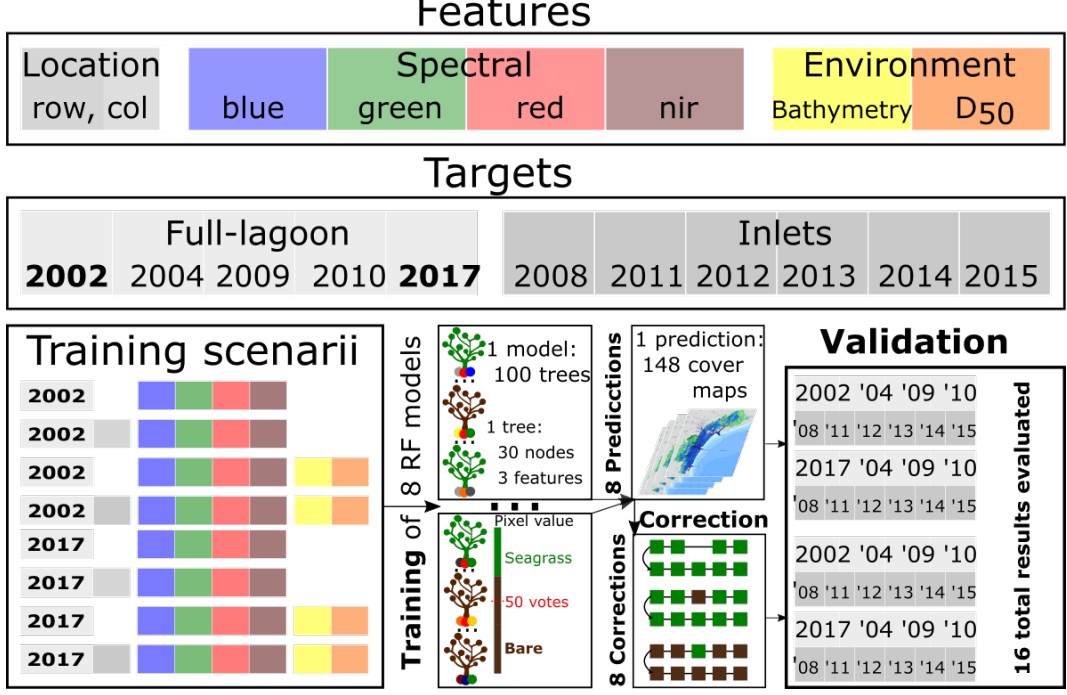

**Figure 5.** Workflow of the study, depicting the different features used (top), the target surveys and the year in which they were performed
(centre); at the bottom, the successive training, prediction, correction and validation steps of the models are described.





## 2.5 Definition of model performance

For binary classifications such as the one performed here, model performance is quantified by use of the numbers of True Positives ($TP$), True Negatives ($TN$), False Positives ($FP$) and False Negative ($FN$).

A global accuracy metric is defined as the ratio of true classifications ($TP+TN$) over the total number of pixels, a ratio often referred to as Overall Accuracy in non-binary classification problems ($OA$, see Table 1). In the Venice Lagoon, the presence of large areas of bare tidal flat relative to seagrass meadow area implies that $OA$ is biased toward the correct prediction of bare tidal flats, rather than toward the correct prediction of seagrass meadows (having a much smaller total area). Hence, Sensitivity ($S$) (Equation 2), and Precision ($P$)(Equation 3), give a measure of the model's performance that better fits our purpose, by eliminating $TN$:

$$S = \frac{TP}{TP+FN} \tag{2}$$

$$P = \frac{TP}{TP+FP} \tag{3}$$

$S$ can be construed as a measure of the model's capacity to identify existing seagrass meadows. In non-binary problems such as those described in Table 1, $S$ becomes User's Accuracy ($UA$) and is specific to each class. Conversely, $P$ reflects the model's capacity not to include bare seafloor in the detected seagrass meadows, and becomes Producer's Accuracy ($PA$) in non-binary problems. These metrics may be combined through their harmonic mean $F1$ (Dice, 1945; Sorensen, 1948), defined in Equation 4:

$$F1 = 2 \cdot \frac{S \cdot P}{S+P} \tag{4}$$

The F1 score increases non-linearly but symmetrically with $S$ and $P$, allowing us to measure the model's performance through a single metric. Contrary to the global accuracy, F1 does not include the term $TN$, and is therefore not affected by the presence of extensive areas unequivocally classified as tidal flats: as such, this metric is more sensitive to changes in the detection of positives. The F1 score is used in the results section to describe the performance of the classifier, which we test on full-lagoon and inlet surveys as shown in Figure 4.

## 3 RESULTS

## 3.1 Effect of added features

In this section, we describe the performance of the RF classifiers in all 8 training scenarios before applying the time-based correction. Figure 6 shows this performance expressed as the F1 score.



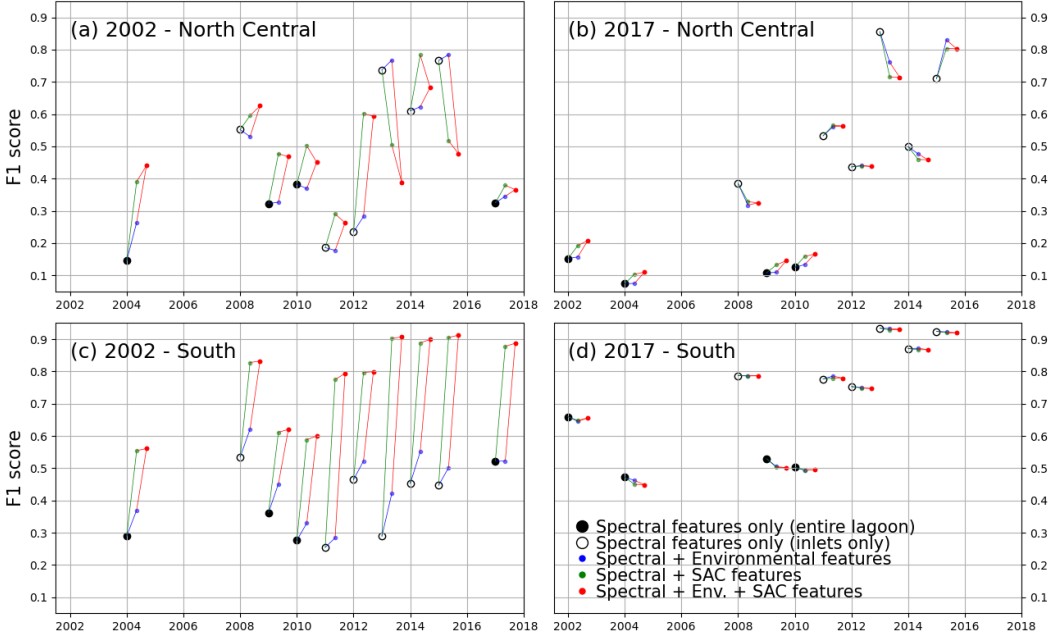

**Figure 6.** F1 score of all models trained, tested on lagoon (full circles) and inlet (empty circles) surveys performed on the years indicated on the x-axis. Black markers indicate models trained only with spectral features. The models tested were trained in the North-Central zone for the 2002 (a) and 2017 (b) surveys, and in the Southern zone for the 2002 (c) and 2017 (d) surveys.

Figure 6 highlights the different behaviours of models trained with the 2002 survey with respect to those trained with the 2017 survey. Models using only spectral features (black) perform very differently than those including either environment (blue, red) or coordinate (green, red) features when trained on a 2002 survey. This is not the case if trained on the 2017 survey. Indeed, while the addition of extra features to a 2002-trained model has a positive effect on the F1 score for most test instances (with exceptions in the North-Central zone), with the addition of location features having the greater effect, it has little, if slightly negative, influence on most 2017-trained models. This indicates that the inclusion among the features of spatial position and environmental features produces large potential benefits, without generating significant negative effects on performance. Such advantages seem to be consistent as the same model formulation is tested across different remote sensing scenes. Decomposing this influence into the effects of Sensitivity and Precision (Figures A2 and A3), we note that improvements occur mostly on the former, while the increase in precision is small (for 2002-trained models) or even negative (for 2017-trained models). In general, Figures A2 and A3 show that 2002-trained models generally have higher precision but lower sensitivity values than 2017-trained models, indicating a tendency of the former to 'miss' vegetated areas while the latter will tend to overestimate the extent of seagrass meadows. In the Southern zone, where most seagrass meadows are located, 2017-trained models using only spectral features perform significantly better than 2002-trained models. 2017-trained models also perform consistently better when tested on inlet surveys rather than lagoon surveys, suggesting that false predictions lie outside the inlet areas. Conversely, 2002-trained models in the Southern zone benefit the most from the addition of environment and coordinate features, improving



F1 scores from under $0.5$ to up to $0.9$. Because the North-Central zone harbours much less extensive seagrass than the Southern
zone, F1 scores are more difficult to evaluate in this region.

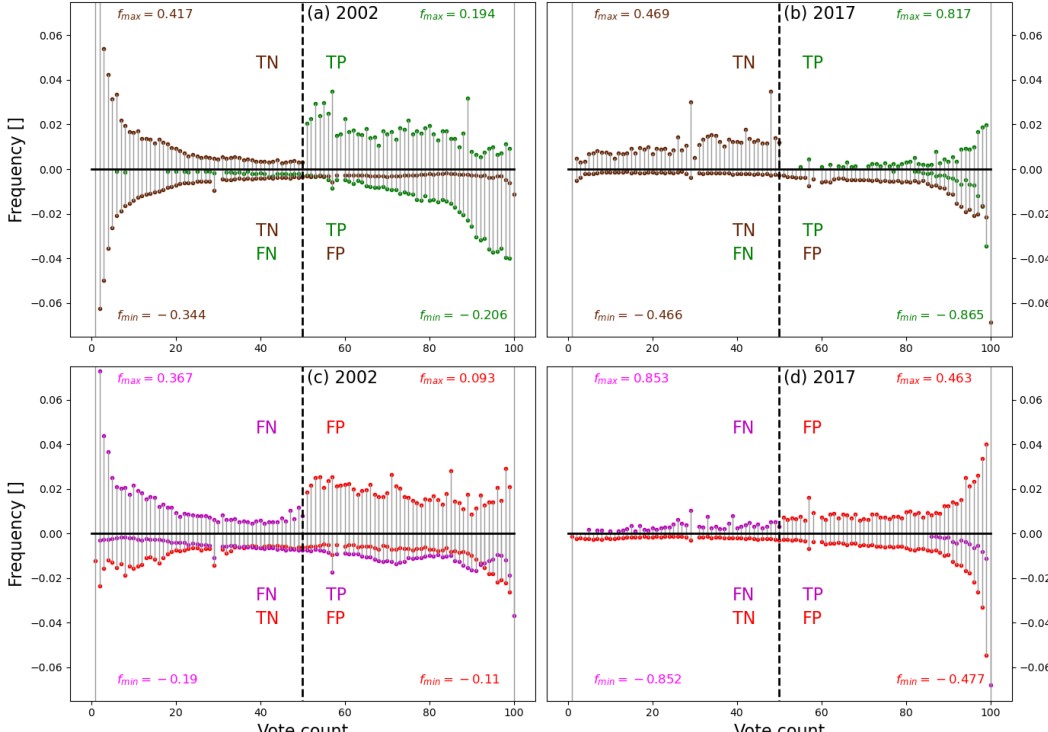

**Figure 7.** Distribution of vote count, i.e. probability to be classified as seagrass by the models in the Southern lagoon, for pixels of different
classifications (vote count < 50 = bare; vote count >= 50 = seagrass). Panels show the summed distributions over all full-lagoon test dates.
Top half of each panel shows classification of pixels with only spectral features, while bottom panel shows the new classification of the same
pixels when also accounting for location features. (a,b) True classifications for 2002-trained and 2017-trained models, respectively; (c,d)
False classifications.

Figure 7 demonstrates in more detail the effect of location features on model behaviour, focusing on the Southern lagoon. It
shows the distribution of correctly classified pixels (a,b) and incorrectly classified pixels (c,d), according to vote count, where
True Positives occur if more than 50 of the 100 trees 'voted' to classify pixels as vegetated.

The top half of each panel shows classifications performed using only spectral features, whereas the bottom half of each
panel shows the distribution of vote count for the same pixels, but using location features as well. For example, all green bars
in the top half are True Positives when classified only with spectral features, but some pixels become False Negatives when
adding location features, hence their positioning on both sides of the 50 vote count separation line in the bottom half of the
panel. Bars corresponding to 0 and 100 in vote count have frequencies much larger than subplot bounds; their values are noted
in the corresponding color for ease of reading.





For 2002-trained models, True Negatives distribution exponentially decreases with vote count, regardless of the inclusion of location features. There are, however, distinctly fewer pixels that are classified with a vote count of 0 in the latter case: this stems from a marginal conversion of these pixels into False Positives. This observation may be repeated for 2017-trained models, with the conversion of True Negative pixels into False Positives being notably more common, with the difference that True Negatives are not exponentially distributed with only spectral features. For True Positives, the proportion of pixels

classified with extreme vote counts(i.e. furthest from the 50 votes line) amounts to only 20% for 2002-trained models, whereas for 2017-trained models it reaches more than 80%. In the latter case, adding location features overall increases the extreme vote counte classifications, which translates into a decrease of classification uncertainty. This shift is even more pronounced for 2002-models, where predictions using only spectral features had a high proportion of low certainty classifications (i.e. closer to the 50 votes line). False Positive classifications for 2017-trained models follow an exponential distribution both

with and without location features. The observed overestimation of seagrass extent (Figure 4) likely prevents the addition of location features from having a significant effect on distributions. On the contrary, 2002 distributions are more variable: the maximum frequency vote count accounts for less than 10% of False Positives and less than 40% of False Negatives when not accounting for location, suggesting that incorrectly classified pixels are more equivocally classified than correct ones. When including location features, many False Positives become True Positives, and vice versa. This accounts for the strong increase

in performance experienced by models trained on 2002 data when accounting for location (Figure 4(c)).

### 3.2   Effect of time-based correction

*Figure 8 examines the effect of the time-based correction process described in Section 2.4 on the F1 score of all tested models.*





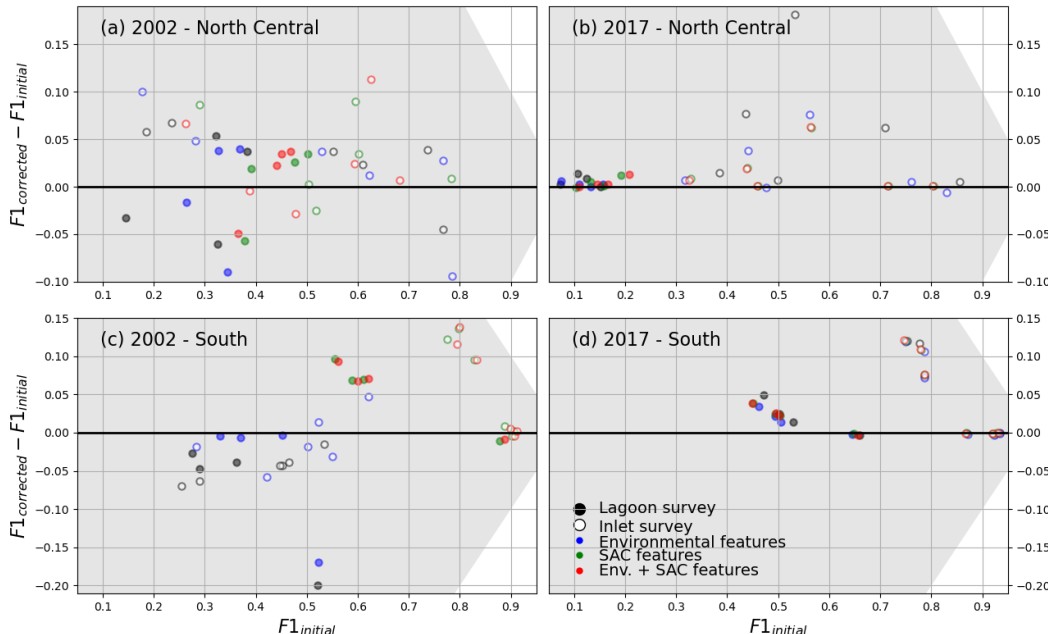

**Figure 8.** Comparison of F1 scores before and after time-based correction (y-axis), tested on lagoon (full circles) and inlet (empty circles) surveys, versus the uncorrected F1 scores (x-axis). Black markers indicate models trained only with spectral features. The models tested were trained in the North-Central zone for the 2002 (a) and 2017 (b) surveys, and in the Southern zone for the 2002 (c) and 2017 (d) surveys. Greyed areas correspond to the boundaries of possible values.

We first observe that while 2002-trained models (a, c) see increases and decreases of similar magnitudes in their F1 scores after correction, 2017-trained models (b, d) mainly see improvements. In the North-Central zone (a, b), no obvious pattern in

the the change in F1 after correction is detected. In the Southern zone (c, d), however, both 2002-trained models and 2017-trained models respond in a discernible pattern to the application of the time-based correction: in 2002-trained models (c), models using only spectral or spectral and environmental features generally see a decrease in F1 after correction, whether they are tested on the full lagoon or at the inlets. This coincides with the lower F1 scores for these models predictions. Conversely, models including known seagrass location as features, for which predictions initially have a higher F1 score, mostly see an

increase of F1 scores. For these models, it appears that the F1 score after time-based correction increases with the initial F1 score. The opposite appears to be true for 2017-trained models (d): different initial F1 scores when testing on lagoon or inlet surveys indicate, within each testing footprint, a decrease of the corrected F1 score with the initial F1 score, while maintaining positive values for most surveys. Notably, whether for lagoon or inlet surveys, tests conducted during the period of the Landsat 7 SLC failure display lower F1 values (see Figure 6(d)) as well as the largest positive change in F1 score after correction: this

suggests that the proportion of true positives reinstated through the time-based correction outweighs the proportion of false predictions added in the process.





## 4 Discussion

Several key issues pertaining to the reliability of seagrass detection arise from the examination of the performance of the models in Section 3: first, the strong disparity in performance, expressed through the F1 score, between models trained using
data from 2002 and 2017; then, the significant difference in the influence of environmental and, particularly, spatial features on F1 scores; finally, the opposing effects of time-based correction on the performance of different models. The behaviour of RF models being non-linear, we do not presume to identify a general relationship between the observations made in a single case study and the performance of seagrass cover predictions using a RF classifier. Nevertheless, we attempt to break down the influence of additional features and time-based corrections, using the Southern zone, where most seagrass is found, as a testing
ground. Figure 9 shows the relative importance of features used in each RF classifier used, where location features dominate, when present, for both 2002-trained and 2017-trained models. In all scenarios, the red band is the most important spectral band for 2017-trained models. As this band is particularly sensitive to suspended sediment concentration, we may surmise that its high importance could impede classification performance, particularly if turbidity is not accounted for in the classification. Conversely, the near-infrared band shares the highest importance among spectral features with the green band. This disparity
may contribute to the diverging behaviours of the models trained on 2002 or 2017 data.

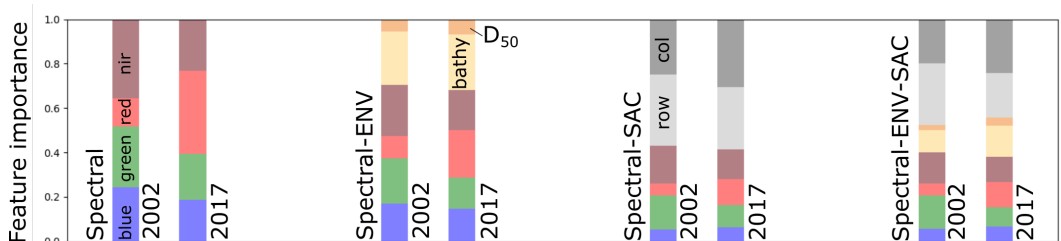

**Figure 9.** Relative importance of each feature used in models trained with 2002 and 2017 data

In Section 1, we mentioned the difficulty of producing consistent seagrass cover predictions at a high frequency. Figure 10 showcases the potential use of location and environmental features to improve the reliability of seagrass cover predictions, demonstrated on a 2002-trained model. Panels (a-d) are each a map of the success rate of seagrass cover predictions for each pixel near Chioggia Inlet, shown in yellow in Figure A1, tested on 36 reference datasets from surveyed and digitised seagrass
cover. In each panel, $R$ represents the average success rate. In dark red, panels (a-d) show the outlines of vegetated patches in the summer 2002 survey. In panels (a,b), where only spectral features were used, we observe that pixels with lower prediction success rates overlap with those with higher occurrence rates, but not necessarily with the presence of seagrass in the summer 2002 survey. Panels (c,d), both of which include location features as well as environmental features, experience notable increase in prediction success rates in these regions. Notably, increases in success rate match areas of high $\Pi$ value (see Figure 4(d)),
suggesting that where spectral features only did not detect the presence of seagrass, models with all features did. Furthermore, we note that the implementation of the time-based correction shown has little effect on R when using only spectral features




(b) but causes an increase when other features are used. This is consistent with our previous observation that the time-based correction improves effective predictions.

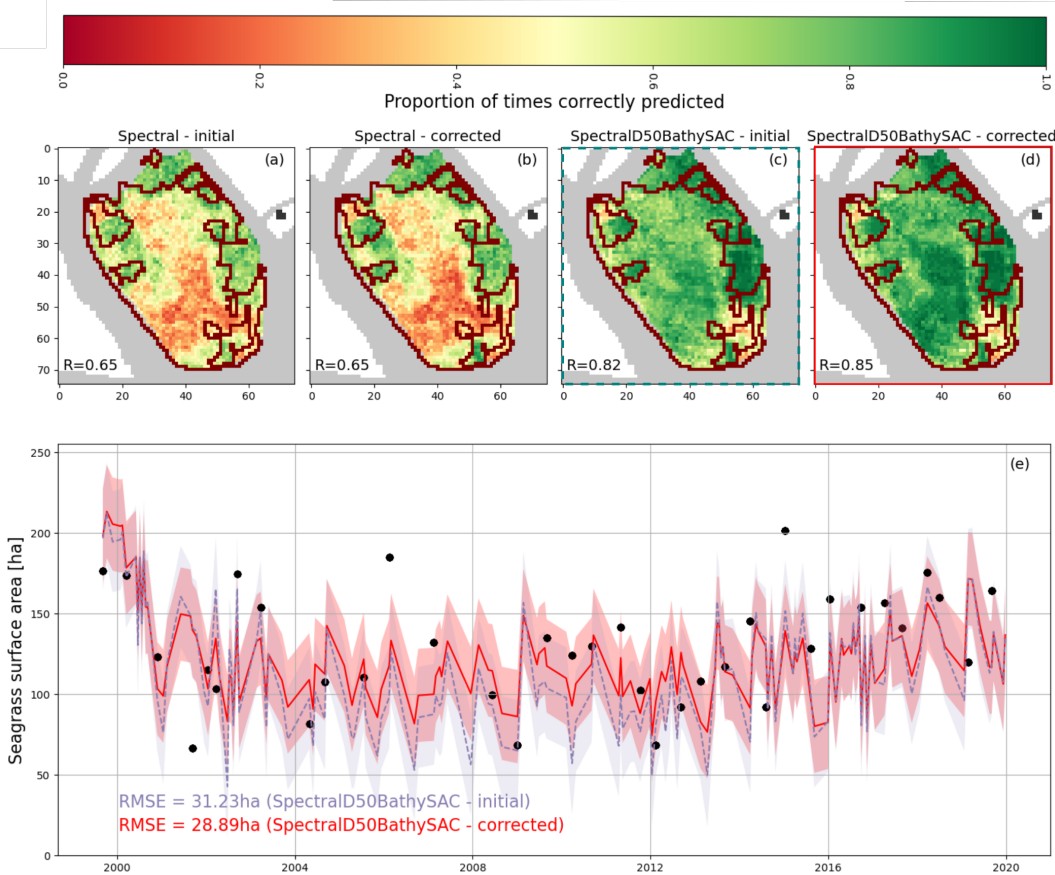

**Figure 10.** (a-d) Map of the proportion of surveys correctly identified for 36 surveys and digitisations combined. R represents the average value across all mapped pixels. Red outlines show the extent of seagrass patches in the summer 2002 survey; (e) Surface area of seagrass predicted by surveys framed in (c,d), filled with the RMSE value. Black dots indicate surveyed or digitised surface area value.

In panel (e), we further note that seagrass surface area predictions decrease by up to a factor of 2 in the year 2000, suggesting
that location features are not demonstrably impeding the model's capacity to predict seagrass change. The root mean square error (RMSE) after implementing the time-based correction is 28.89ha, corresponding to a 7.5% decrease in RMSE respective to predictions without the time-based correction and amounting to 22% of the average seagrass surface area in the examined pixels. This value matches and exceeds half of the variation in seagrass surface area between most dates where vegetation cover is predicted. While the error in surface area prediction remains high for the purposes of monitoring high frequency changes in
seagrass cover, it represents a significant improvement over models using only spectral features and no time-based corrections.





# 5   Conclusions

In this contribution, we implemented several random forest classifiers using various sets of features, supplemented by time-based corrections, to detect the spatial and temporal dynamics of seagrass meadows in the Venice Lagoon. We trained 8 such models using a combination of spectral data, bathymetry, median sediment grain size, and coordinates of known seagrass extent

as features. 4 models were trained using data taken from field surveys and Landsat images of summer 2002, and 4 others with data taken in 2017.

We found that 2002-trained models and 2017- trained models responded differently to both the addition of out-of-image features and to the time-based correction. Adding location information was shown to significantly improve the $F1$ score without preventing the model from detecting variations in seagrass cover, but only for 2002-trained models. Examining the

importance of different features in each of the models, we observed that 2002-trained models and 2017-trained models were under the dominating influence of seagrass location, when used, but otherwise did not share the same most important spectral features. This may be a reason for their discordant behaviour. Furthermore, the vote count, which expresses the probability of a given pixel to be seagrass according to a RF model, was generally more polarised toward extreme values in 2017-trained models. While not a root cause of the difference between the sets of models, it revealed the effect of using geographic location

features on tree voting. This shows that true change in seagrass meadows can be detected by location-dependant models when taking into account prior knowledge of seagrass location. Ultimately, accurately accounting for spatial auto-correlation will require the examination of more cases as well as a generally applicable formulation of its influence.

*This contribution demonstrated the potential of mixing spatial auto-correlation patterns and time-series analysis with commonly used spatial detection methods, such as Random Forest classifiers to increase the performance of submerged vegetation*

*detection methods. The positive effects of these modifications to standard methods have been shown to outweigh their potential "side-effects", although further research is required to generalize their influence. The resulting increased confidence in each seagrass cover map allows the generation of dense sequences of maps, through which the space-time dynamics of seagrass meadows may be described and connected to their governing environmental drivers. This, in turn, can inform ecological models, sediment transport simulations and ultimately feed into predictions of tidal basin response to environmental change.*

*Data availability.* The satellite data used in this contribution are openly available on multiple platforms, including NASA's EarthExplorer (https://earthexplorer.usgs.gov/). The data concerning seagrass cover surveys, bathymetry and median sediment grain size are openly available via the Atlante della Laguna at http://cigno.atlantedellalaguna.it/maps. Tide and wind gauge data are openly available at https://www.comune.venezia.it/content/dati-dalle-stazioni-rilevamento.



**Appendix A**



**Figure A1.** Footprint of the inlet surveys of Lido, Malamocco and Chioggia (blues) and of the digitised patch (yellow). The coordinate system used is EPSG: 3003. Inset image: ©Google Earth



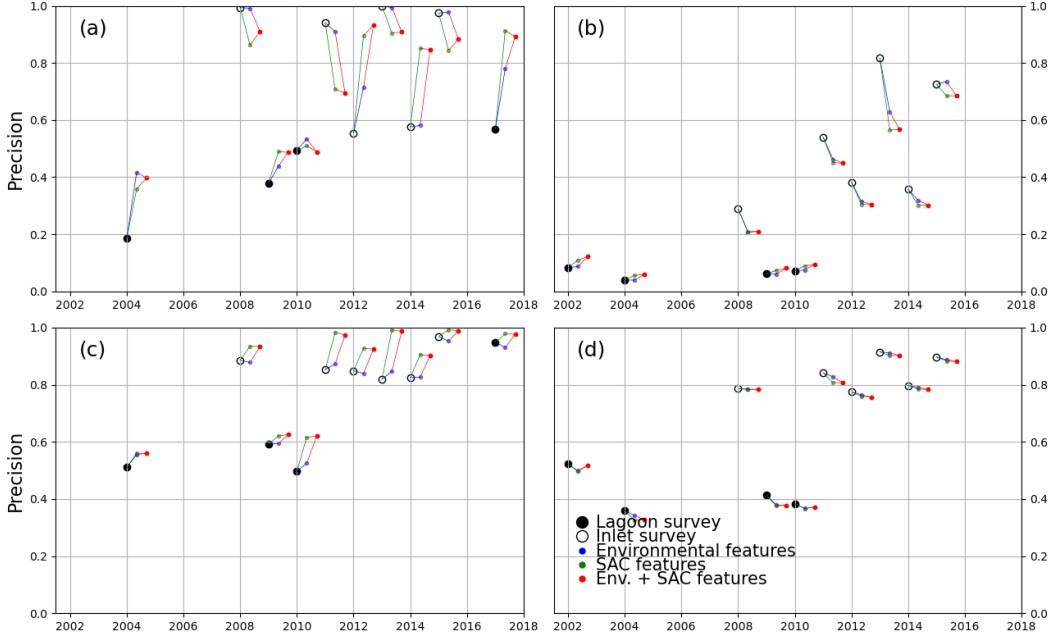

**Figure A2.** Precision score of all models trained, tested on lagoon (full circles) and inlet (empty circles) surveys performed on the years indicated on the x-axis. Black markers indicate models trained only with spectral features. The models tested were trained in the North-Central zone for the 2002 (a) and 2017 (b) surveys, and in the Southern zone for the 2002 (c) and 2017 (d) surveys.





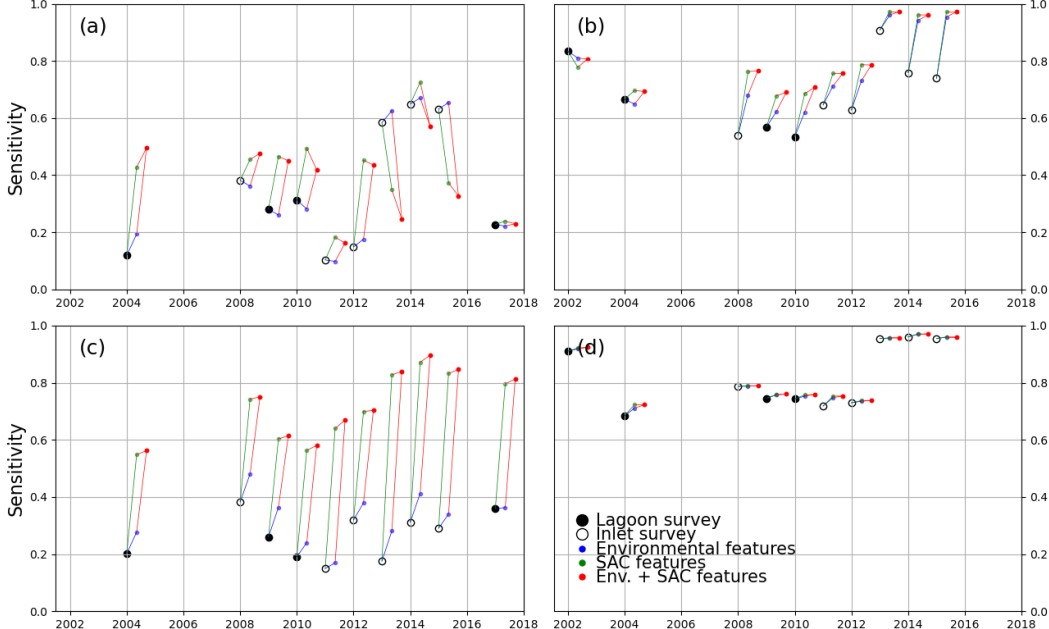

**Figure A3.** Sensitivity score of all models trained, tested on lagoon (full circles) and inlet (empty circles) surveys performed on the years indicated on the x-axis. Black markers indicate models trained only with spectral features. The models tested were trained in the North-Central zone for the 2002 (a) and 2017 (b) surveys, and in the Southern zone for the 2002 (c) and 2017 (d) surveys.

-0cm





**Table A1.** Compilation of published works on seagrass detection from satellite data

| Reference number | Reference |
|:---:|:---:|
| 1 | Phinn et al. (2008) |
| 2 | Pu et al. (2012) |
| 3 | Lyons et al. (2012) |
| 4 | Dekker et al. (2005) |
| 5 | Wabnitz et al. (2008) |
| 6 | Hossain et al. (2015) |
| 7 | Misbari and Hashim (2016) |
| 8 | Topouzelis et al. (2018) |
| 9 | Kovacs et al. (2018) |
| 10 | Kohlus et al. (2020) |
| 11 | Pu et al. (2012) |
| 12 | Pu et al. (2012) |
| 13 | Kovacs et al. (2018) |
| 14 | Zoffoli et al. (2020) |
| 15 | Traganos et al. (2018); Traganos and Reinartz (2018) |
| 16 | Kovacs et al. (2018) |
| 17 | Phinn et al. (2008) |
| 18 | Roelfsema et al. (2014) |
| 19 | Phinn et al. (2008) |
| 20 | Bakirman and Gumusay (2020) |
| 21 | Kovacs et al. (2018) |
| 22 | O'Neill and Costa (2013) |
| 23 | Amran (2017) |

*Author contributions.* Conceptualization, G. Goodwin, L. Carniello, A. D'Alpaos, M. Marani and S. Silvestri; methodology, G. Goodwin, L. Carniello, A. D'Alpaos, M. Marani and S. Silvestri; software, G. Goodwin.; validation, G. Goodwin, L. Carniello, A. D'Alpaos, M. Marani and S. Silvestri; formal analysis, G. Goodwin; investigation, G. Goodwin; resources, M. Marani; data curation, G. Goodwin; writing—original draft preparation, G. Goodwin; writing—review and editing, G. Goodwin, L. Carniello, A. D'Alpaos, M. Marani and S. Silvestri; visualization, G. Goodwin.; supervision, L. Carniello, A. D'Alpaos, M. Marani and S. Silvestri; project administration, M. Marani; funding acquisition, M. Marani. All authors have read and agreed to the published version of the manuscript.



**Table A2.** Selected Landsat scenes with full-lagoon or inlet surveys

| Lagoon surveys | LE07_L1TP_192028_20020914_20170128_01_T1 |
|---|---|
| | LE07_L1TP_192028_20040903_20170119_01_T1 |
| | LE07_L1TP_192028_20090901_20161218_01_T1 |
| | LE07_L1TP_192028_20100904_20161212_01_T1 |
| | LC08_L1TP_192028_20170830_20170914_01_T1 |
| Inlet surveys | LE07_L1TP_192028_20080610_20161228_01_T1 |
| | LE07_L1TP_192028_20090901_20161218_01_T1 |
| | LE07_L1TP_192028_20100904_20161212_01_T1 |
| | LE07_L1TP_192028_20111009_20161206_01_T1 |
| | LE07_L1TP_192028_20120909_20161129_01_T1 |
| | LC08_L1TP_192028_20130904_20170502_01_T1 |
| | LC08_L1TP_192028_20140806_20170420_01_T1 |
| | LC08_L1TP_192028_20150809_20170406_01_T1 |

*Competing interests.* Some authors are members of the editorial board of the current special issue "Monitoring coastal wetlands and the seashore with a multi-sensor approach". The peer-review process was guided by an independent editor, and the authors have also no other competing interests to declare.

*Acknowledgements.* This research was funded by CORILA under the project Venezia 2021 (grant CUP D51B02000050001).



**Table A3.** Selected Landsat scenes with digitised areas

| Digitised scenes | LE07_L1TP_192028_19990906_20170217_01_T1 |
|---|---|
| | LE07_L1TP_192028_20000316_20170213_01_T1 |
| | LE07_L1TP_192028_20001127_20170209_01_T1 |
| | LE07_L1TP_192028_20010911_20170203_01_T1 |
| | LE07_L1TP_192028_20020117_20170201_01_T1 |
| | LE07_L1TP_192028_20020322_20191106_01_T1 |
| | LE07_L1TP_192028_20020914_20170128_01_T1 |
| | LE07_L1TP_192028_20030325_20170214_01_T1 |
| | LE07_L1TP_192028_20040428_20170121_01_T1 |
| | LE07_L1TP_192028_20050720_20170113_01_T1 |
| | LE07_L1TP_192028_20060213_20170110_01_T1 |
| | LE07_L1TP_192028_20070216_20170104_01_T1 |
| | LE07_L1TP_192028_20090104_20161223_01_T1 |
| | LE07_L1TP_192028_20100328_20161215_01_T1 |
| | LE07_L1TP_192028_20110502_20161210_01_T1 |
| | LE07_L1TP_192028_20120214_20161203_01_T1 |
| | LE07_L1TP_192028_20130216_20161126_01_T1 |
| | LC08_L1TP_192028_20140331_20170424_01_T1 |
| | LC08_L1TP_192028_20150113_20170414_01_T1 |
| | LC08_L1TP_192028_20160116_20170405_01_T1 |
| | LC08_L1TP_192028_20160928_20170321_01_T1 |
| | LC08_L1TP_192028_20170408_20180523_01_T1 |
| | LC08_L1TP_192028_20170830_20170914_01_T1 |
| | LC08_L1TP_192028_20180326_20180404_01_T1 |
| | LC08_L1TP_192028_20180630_20180716_01_T1 |
| | LC08_L1TP_192028_20190225_20190309_01_T1 |
| | LC08_L1TP_192028_20190905_20190917_01_T1 |

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
