# Peer review of "Toward coherent space-time mapping of seagrass cover from satellite data: example of a Mediterranean lagoon"

_EGUsphere, 2022_

## Referee Comment (RC1)

Title: Toward coherent space-time mapping of seagrass cover from satellite data: example of a Mediterranean lagoon
Author(s): Guillaume Cyril Henri Goodwin et al.
MS No.: egusphere-2022-1501
MS type: Research article
Special Issue: Monitoring coastal wetlands and the seashore with a multi-sensor approach

**General comments**

This study is investigating ways to improve the performance of remote sensing methods to predict the presence seagrass consistently over time. The addition of bathymetry, sediment size and coordinates data are indeed very relevant and deserving of such study.

The introduction section is well documented and written.

However, I find the methods and overall presentation of the results quite confusing and hard to interpret. Furthermore, very little attention is given to the ecological relevance of the findings and their possible application in other areas with different seagrass species and cover. The discussion section needs to include how the finding of the study fit in the seagrass remote sensing literature detailed in the introduction and how change detection can be improved.

I have suggested in the comment below various ways I think the authors can significantly improve the quality of this publication. I would strongly recommend major revisions before publication.

**Specific comments**

| | |
|---|---|
| Lines 105-110 | Please provide some more background information about the field survey data. It is not clear how those maps which will be used as training data were produced in the first place (remote sensing ?). |
| Lines 114,115 | Please provide more information about how and why the specific patch of seagrass near Chioggia was digitized. |
| Line 125 | Please provide explanation for choosing the specific 0.75m threshold. |
| Figure 3 | This figure is very complex to interpret. Consider revising how this information is displayed to only show curated results. |
| Description of random forest section | Could you confirm that the input data for the random forest models are individual pixels. Please provide more information about training/validation/test data split ratios. Also give ratio of pixel classified as 0 and 1 (balanced ?) |
| Definition of model performance section | I am confused and possibly concerned about the method used to compare model performances. The main issue for me is the variability in size of the testing datasets. I'm assuming the full lagoon datasets have more |

| | pixels than the inlet ones therefore making it more prone to errors. This is quite complex and maybe it would be more useful to report the average F1 score for similar testing sets (eg. Full lagoon vs inlet). |
|---|---|
| Figure 6 | Consider updating with the comments from above |
| Results – Effect of added feature | Comparing model performance only based on metrics can be misleading. Indeed a lot of factors can bias those metrics such as sample size and class imbalanced. It would be very useful for the reader to have some actual visual representation (maps) of the prediction vs ground truth. I am suspecting that the Southern zone and Northern zone differences are due to training pixel class imbalance.
To me all the comparisons between the models trained on specific year and zone just highlight the effect of training bias. I suggest instead to conserve the training scenarios based on the different features combination (spectral, location, environment) and pool the 2002 and 2017 training data together. Adding cross validation to your random forest would ensure the results are not influenced by a particular year or area. Consider also doing more fine tuning of the number of trees and nodes. I understand you chose 100 trees to make it easier for the vote count to show as a probability percentage. |
| Figure 7 | I don't think this figure is really that informative and I would suggest to be moved to the appendix. Consider here to summarize better the data from the figure. Try to better identify the clear patterns. What I would be looking for in this section is how does the different feature combination of the training scenarios influences the probabilities of a pixel to be classified as seagrass or bare. More importantly try to identify where those potential misclassifications occur (e.g. near the edges of the meadows), what could be causing it, and how adding features such as location, depth and grain size might improve the predictions. |
| Figure 8 | Again here I found the figure quite hard to read due to the large number of points and them clumping. I would suggest here a boxplot showing the range difference of F1 score before and after correction for the categories represented by the various marker type and colours. |

| Effect of time-based correction | As before judging the effect of the time-based correction solely on F1 score might be misleading. What I would be looking for here is what is the influence of the time-based correction on the seagrass area predicted. Does it create more cohesive meadows by reducing the salt and pepper effect of pixel-based predictions? Does it allow large areas that were not classified previously to get predicted properly now? What are the ecological implications of such differences in predictions |
|---|---|
| Figure 10 | I find this kind of figure much more informative that the ones in the previous result section. This clearly shows the value of the additional features in the RF models combined with the time-based correction. I like also the addition of the effect if the correction on the predicted seagrass area.
This is the most compelling figure so far and I would have like to see more of that sooner in the paper.
I know understand what the digitised patch was for. It would help the reader to explain the purpose of this patch earlier in the method. |
| Line 339-340 | Please expand on that point. This is a very interesting a relevant observation. Show how the improvements to the model are beneficial for detecting change in seagrass area in greater detail. This is the most important implication and needs to be highlighted more. |

Technical comments

| Lines 32,35 | Inconsistent references format |
|---|---|
| Line 36 | Typo to be deleted (-0cm) |

---

## Author Response (AR1)

**Response to Reviewer 1**

We thank both reviewers for their valuable contributions to this article. In light of their comments, we have amended both the main text and some figures, in the hope that these changes will improve the clarity of our article. We particularly thank reviewer 1 for their detailed comments, which enabled us to clarify the visuals and the narrative of the contribution, as well as add important technical precisions.

Below, we respond to the comments from reviewer 1 numbering them by order of appearance in the original manuscript. Where comments have lead directly to a modification of figures or text, the reviewer will find modified sections referenced to their line numbers in the revised manuscript, where modified text has been typefaced in italics for the purpose o f the review.

**Response to individual comments**

R1 C1: Please provide some more background information about the field survey data. It is not clear how those maps which will be used as training data were produced in the first place (remote sensing ?).

We have altered the text in consequence, detailing the methods and objectives of the field surveys. - L109-112

R1 C2: Please provide more information about how and why the specific patch of seagrass near Chioggia was digitized.

The reference to the demonstration of surface area change detection, which the reviewer later noted, has been added for clarity. - L120-122

R1 C3: Please provide explanation for choosing the specific 0.75m threshold.

Imposing the value of 0.75m as a maximum sea level was an arbitrary choice. Here we chose to let the RF work without correction to demonstrate portability of the addition of new features in areas where suspended sediment load is not well known.- L134-137

R1 C4: Could you confirm that the input data for the random forest models are individual pixels. Please provide more information about training/validation/test data split ratios. Also give ratio of pixel classified as 0 and 1 (balanced ?)

We have introduced clarifications in the Description of the RF model, throughout the section. L 215-221

R1 C5:[Figure 3] is very complex to interpret. Consider revising how this information is displayed to only show curated results

We have modified Fig3, caption and interpretation text to diminish the information load. – Figure

R1 C6I am confused and possibly concerned about the method used to compare model performances. The main issue for me is the variability in size of the testing datasets. I'm assuming the full lagoon datasets have more pixels than the inlet ones therefore making it more prone to errors. This is quite complex and maybe it would be more useful to report the average F1 score for similar testing sets (eg. Full lagoon vs inlet).

We acknowledge the reviewer's observation that different sizes of datasets lead to a different assessment of error. In the particular case of the Venice Lagoon, large continuous surfaces of the lagoon (mainly in the central western area) are devoid of seagrass and are also never classified as vegetated by the model, mitigating in an unquantified manner the error caused by a larger dataset. Because of this, the global accuracy metric, which is dependant on the quantity of true negatives, is biased toward higher values, hence our use of the F1 score to reduce the bias. While Figure 6 does distinguish between full lagoon surveys and inlet surveys, it is true that we did not comment on this aspect. We have endeavoured to stress different interpretations of the performance for different areas throughout the text of the figure analysis. - Figure 6 interpretation text was modified in consequence. - L 270-277

RA C7:  Comparing model performance only based on metrics can be misleading. Indeed a lot of factors can bias those metrics such as sample size and class imbalanced. It would be very useful for the reader to have some actual visual representation (maps) of the prediction vs ground truth. I am suspecting that the Southern zone and Northern zone differences are due to training pixel class imbalance. To me all the comparisons between the models trained on specific year and zone just highlight the effect of training bias. I suggest instead to conserve the training scenarios based on the different features combination (spectral, location, environment) and pool the 2002 and 2017 training data together. Adding cross validation to your random forest would ensure the results are not influenced by a particular year or area. Consider also doing more fine tuning of the number of trees

and  odes. I understand you chose 100 trees to make it easier for the vote count to show as a probability percentage

We agree with the reviewer that our original presentation of results was misleading. We have substantially modified both the figures and the interpretation of results to match them (see below). - L296-319

R1 C8: I don't think this figure is really that informative and I would suggest to be moved to the appendix. Consider here to summarize better the data from the figure. Try to better identify the clear patterns. What I would be looking for in this section is how does the  different feature combination of the training scenarios influences the probabilities of a pixel to be classified as seagrass or bare. More importantly try to identify where those potential misclassifications occur (e.g. near the edges of the meadows), what could be causing it, and how adding features such as location, depth and grain size might improve the predictions

This figure was deleted in favour of displaying maps of correct prediction rates. A similar map is also added when examining the effect of the time-based correction. Figure 7 => Figures 7-9

R1 C9: Figure 8: Again here I found the figure quite hard to read due to the large number of points and them clumping. I would suggest here a boxplot showing the range difference of F1 score before and after correction for the categories represented by the various marker type and colours.

The number of points examined in each category is too low to justify a boxplot, but we have modified the figure to reduce clumping and highlight the different behaviour between lagoon and inlet surveys. - Figure 10

R1 C10: As before judging the effect of the time-based correction solely on F1 score might be misleading. What I would be looking for here is what is the influence of the time-based correction on the seagrass area predicted. Does it create more cohesive meadows by reducing the salt and pepper effect of pixel-based predictions? Does it allow large areas that were not classified previously to get predicted properly now? What are the ecological implications of such differences in predictions

We thank the reviewer for their insight, and again have added a map of correct prediction rates to help visualise the model results in a condensed manner. We have

also added explanatory text as to the consequences of the time-based correction for practical applications. - Section 3.2

R1 C11:  I find this kind of figure much more informative that the ones in the previous result section. This clearly shows the value of the additional features in the RF models combined with the time-based correction. I like also the addition of the effect if the correction on the predicted seagrass area. This is the most compelling figure so far and I would have like to see more of that sooner in the paper. I know understand what the digitised patch was for. It would help the reader to explain the purpose of this patch earlier in the method.

We have attempted to introduce the reader to this application earlier on in the article.

R1 C12: Line 339-340 : Please expand on that point. This is a very interesting a relevant observation. Show how the improvements to the model are beneficial for detecting change in seagrass area in greater detail. This is the most important implication and needs to be highlighted more

An expanded discussion was added. L381-390

**Response to Reviewer 2**

We thank both reviewers for their valuable contributions to this article. In light of their comments, we have amended both the main text and some figures, in the hope that these changes will improve the clarity of our article. We particularly thank reviewer 2 for their cautious reading of our manuscript, which enabled us to change both the wording of problematic statements and add necessary technical details.

Below, we respond to the comments from reviewer 2 numbering them by order of appearance in the original manuscript. Where comments have lead directly to a modification of figures or text, the reviewer will find modified sections referenced to their line numbers in the revised manuscript, where modified text has been typefaced in italics for the purpose of the review.

R2 C1: The authors justify why they chose RF with Table 1 but it is not clear in the introduction what is meant by "modified" RF? Even though this modified RF is served as the novelty of the manuscript, from what I see authors propose a set features and

a time-based correction rather than a modification within in the method itself. This should be clarified.

 Here we changed the text to avoid leading the reader to believe that we designed a novel method. - L75-80

R2 C2:This also includes a better literature survey since authors seem to overlook to mention about the comprehensive review about seagrass mapping with satellite remote sensing by Hossain et al. (2015) https://doi.org/10.1080/01431161.2014.990649.

We thank the reviewer for this additional reference, which should not have been overlooked. We have referenced this source and commented on its implications for our paper.  - L72-75

R2 C3: The specific species of the seagrass can be given. Posidonia, Cymodoceaceae, Zosteraceae? Etc.

The species were added in the description of the surveys. - L109-112

R2 C4: Lines 120-125: "In May 2003, the Scan Line Correction (SLC) system on the Landsat 7 ETM sensor failed, causing all subsequent ETM scenes to contain strips of empty data. Nevertheless, we did not disregard these acquisitions, and instead have designed our methods to account for missing data" What methods?

Here we changed the text to avoid leading the reader to believe that we designed a novel method rather than demonstrated potential of using additional features. - L130-131

R2 C5: Lines 125-135: From the text, I understand that the authors do not apply water column correction to the images. From my point of view, this can create problems in areas where albedo changes with different seabed features and depth. I think this should be clarified and justified in more detail.

Imposing the value of 0.75m as a maximum sea level was an arbitrary choice. Here we chose to let the RF work without correction to demonstrate portability of the addition of new features in areas where suspended sediment load is not well known.- L134-137

R2 C6:2/ I think the authors used RGB and NIR bands, it should be stated more clearly. Moreover, even though wavelengths of the RGB bands in two satellite mission are quite similar, the wavelength of the NIR band is quite different (0.77-0.90 in Landsat 7; 0.85-0.88 in Landsat 8). Maybe this should be discussed in the relevant section.

The paragraph was modified to account for this comment. - L162-178

R2 C7: My final comment will be about the set of features. I advise authors to also try the RGB band combination since NIR band can cause disturbance due to possible Chl-a concentration in the area.

We agree with the reviewer that considering RGB band combinations often improves detection quality when Chl-a in the water column reduces contrast between bare soil and vegetated areas in the NIR band. In this particular case however, we did test the results when excluding the NIR band specifically and found the performance to be lesser than when including this band. This information has been added to the manuscript. - L356-360